# Physiological Characteristics of Field Bean Seeds (*Vicia faba* var. *minor*) Subjected to 30 Years of Storage

Agnieszka I. Piotrowicz-Cieślak [1],*, Magdalena Krupka [1], Dariusz J. Michalczyk [1], Bogdan Smyk [2], Hanna Grajek [2], Wiesław Podyma [3] and Katarzyna Głowacka [1]

1   Department of Plant Physiology, Genetics and Biotechnology, Faculty of Biology and Biotechnology, University of Warmia and Mazury in Olsztyn Oczapowskiego 1A, 10-718 Olsztyn, Poland; magkrup1209@gmail.com (M.K.); darim@uwm.edu.pl (D.J.M.); kasiag@uwm.edu.pl (K.G.)
2   Department of Physics and Biophysics, Faculty of Food Science, University of Warmia and Mazury in Olsztyn, Oczapowskiego 4, 10-719 Olsztyn, Poland; bsmyk@uwm.edu.pl (B.S.); grajek@uwm.edu.pl (H.G.)
3   Plant Breeding and Acclimatization Institute (IHAR), National Research Institute, Radzików, 05-870 Błonie, Poland; w.podyma@ihar.edu.pl
*   Correspondence: acieslak@uwm.edu.pl; Tel.: +48-89-5234289

**Abstract:** Seed vigour and viability, synchronous fluorescence spectroscopy, and proteomic profiles were analysed in field bean (Vicia faba var. minor) (*Vicia faba* var. *minor*) seeds (two cultivars) subjected to dry storage at −14 °C or +20 °C for 30 years. The seeds stored at −14 °C retained very high germinability (91–98%) until the end of the experiments, while seeds from the same lots but stored at room temperature completely lost viability. The deterioration of seeds stored at +20 °C was also manifested by a vast (4- to 6-fold) increase in leachate electroconductivity, and the changes in synchronous spectra and proteomic profiles. To carry out detailed analyses of seed proteins, protein extracts were pre-purified and divided into albumin, vicilin, and legumin. Only one protein, superoxide dismutase, was more abundant in deteriorated seeds (of one cultivar) compared to the high vigour seeds. The results show that seed deterioration strongly and specifically affects the contents of some storage proteins. Moreover, the colour of seed coats changes gradually, and seeds stored at −14 °C were light brown, while those constantly exposed to +20 °C turned black. Synchronous fluorescence spectroscopy showed that this change of colour was caused by formation of oxidized and condensed phenols and that the phenol content in seed coats decreased parallel to seed deterioration.

**Keywords:** albumin; legumin; long-term storage; *Vicia faba* var. *minor* seeds; vicilin; testae; synchronous fluorescence spectra

## 1. Introduction

Provision and storage of crops are vital tasks of agriculture. Seed lots with particular economic value are stored in seed banks where they can be preserved for long periods [1]. Field (*Vicia faba* var. *minor*) seeds are considered valuable crops as they rank the seventh most important protein source among grain legumes, worldwide [2]. They are used as animal feed, rather than food for humans.

The rate of seed ageing strongly depends on seed water content, in addition it is also determined by the presence of microflora, seeds maturity, the relative humidity of ambient air, oxygen partial pressure, storage temperature, and other factors. Considering seed storability and its physiological/genetic determinants, seeds have been divided into three groups: orthodox, i.e., desiccation-tolerant, recalcitrant (damaged by desiccation and characterised by low storability), and intermediate (partially tolerant to desiccation but sensitive to low temperature) [3]. The orthodox seeds survive dehydration to a very low water content (5–10% dry mass) and their storability increases parallel to the decrease in

water content and storage temperature [4]. Seeds of most crops, including field bean (*Vicia faba* var. *minor*), belong to the orthodox category. According to the international guidelines, they should be stored in air-tight containers at the temperature from −20 °C to −15 °C and low relative humidity [5]. Recalcitrant seeds cannot be successfully stored under the same regime. Cryopreservation in liquid nitrogen is the recommended storage procedure in this case [4].

Even with the most careful adherence to the prescribed rules of long term storage, a gradual deterioration of seed, resulting from seed ageing processes, will always occur [6]. Decreasing germination rate and increased frequency of malformed seedlings are the first symptoms of seed ageing [1]. The causes of seed ageing and eventually death are not fully understood. Several hypotheses have been proposed to explain this process, for instance, starvation of meristematic cells, genetic degradation (somatic mutations), the action of hydrolytic enzymes, enzyme degradation and inactivation of ribosomes, and accumulation of toxic compounds and free radicals [7]. Free radical reactions are considered the main cause, as was suggested by Harman [8]. According to this model, reactive oxygen species (ROS) accumulating as a result of spontaneous reactions occurring in stored seeds, oxidize lipids of cell and mitochondrial membranes, impair DNA, RNA, and proteins and initiate the Maillard reactions [9]. Moreover, the transition of cytoplasm from the glassy to liquid state takes place and it favours protein denaturation and irreversible aggregation [10].

The damages to proteins, resulting from the accumulation of ROS, may lead to decreased seed vigour and viability [6,9]. The relation of proteins to seed storability seems particularly clear in the case of late embryogenesis abundant proteins (LEA), that are involved in maintaining the glassy state of the cytoplasm, and some enzymes, involved in scavenging of the free radicals [1]. The effect of the state of storage proteins on seed storability has not been so obvious so far. It was clear however, that the storage proteins in addition to their main role, may also participate in defence mechanisms protecting seed cells from phytopathogens [11]. Recent studies by Nguyen et al. [12] suggest that storage proteins contribute to the maintenance of seed vigour and viability. Considering the abundance of storage proteins and their high preponderance to oxidation, it was postulated that they form a very important component of the intracellular ROS scavenging system. Proteomic analysis of seed proteins may thus turn out to be a sensitive tool for estimation of seed vigour and viability; not without difficulties—high variability of storage proteins in terms of their physical or chemical properties and very high concentrations of some members of this group of molecules make the proteomic analysis of seeds rather challenging. Nevertheless, it was postulated a new seed vigour and viability test could be developed based on such analyses [13]. There is ample literature suggesting the role of oxidative processes in the mechanism of seed deterioration. The targets of such reactions could be diverse, including proteins (storage and enzymatic), membrane lipids, and secondary metabolites, including phenols [14–16].

Analysing the changes occurring in seeds subjected to long-term storage is a task with a great demand on planning and time. Therefore, studies on the relationship between seed physiological or biochemical characteristics and their storability are quite rare. The objective of this paper was to assess vigour and viability of field bean (*Vicia faba* var. *minor*) seeds subjected to 30 years of storage at −14 °C and +20 °C and to relate these characteristics to synchronous fluorescence spectra of seed coat metabolites and the proteomic profile of seed storage proteins.

## 2. Materials and Methods

### 2.1. Material Storage Condition

Seeds of two cultivars "Nadwiślański" and Dino of field bean (*Vicia faba* var. *minor*), obtained from the Plant Breeding and Acclimatization Institute in Radzików, Poland, were used in the experiments, harvested in 1989. The seeds were stored in dark for 30 years in air-tightly closed glass jars at −14 °C and +20 °C temperature and the relative air humidity 46% and 34%, respectively. Next, the seeds were transferred to an ultra-freezer (−86 °C) and were stored there until analyses (for the period of

up to three months). Hair hygrometer (Präzisions Haarhygrometer; Feingerätebau K. Fischer GmbH, Drebach, Germany) was used to measure the air relative humidity. At the onset of the experiments seed water content (determined with the gravimetric, i.e., oven-dry method) was 8.8 ± 0.2%. After 30 years of storage seed water content in Nadwiślański was 10.6% and 6.6%, while in Dino it was 8.6% and 6.5% for seed stored at −14 °C and +20 °C, respectively. The density was determined as the ratio of sample mass to the true volume of the seeds which was determined by displacement method using pycnometer [17].

### 2.2. Vigour and Viability

Vigour and viability assessments (based on seed germination, seedling growth, and seed leachate electroconductivity) were carried out following the ISTA [18] recommendations. For seed germination test 50 seeds were sown on sheets of Anchor Paper (USA) wetted with deionized water. Seed germination and seedling growth was estimated in seven-day-old plants (grown at 12 h/12 h day/night photoperiod and temperatures 20 °C/16 °C at day/night, respectively) based on the average lengths of roots and stems. To determine the electroconductivity of seed leachates 50 seeds were soaked in 250 mL deionized water for 24 h at 20 °C. A HI 2315 conductometer (Hanna Instruments) was used and the results were expressed as $\mu S \times cm^{-1} \times g^{-1}$. Seed germination, seedling growth and seed leachate electroconductivity test were carried out in five independent replications.

### 2.3. Seed Coat Analyses

For scanning electron microscopy (SEM) observations the seeds were coated with gold using a JEOL JFC 1200 ion coater and observed in a JEOL JSM–5310LV (Akishima, Japan) scanning electron microscope at 20 kV.

### 2.4. Isolation of Seed Phenols

The testae were mechanically removed from seeds of both varieties, stored −14 °C and +20 °C for 30 years. They were next pulverized using an electric mill (IKA A11 basic, IKA-WERKE GmbH & Co. Staufen im Breisgau, Germany). Phenols were extracted from the seed coat powders (200 mg samples) with 20 mL 70% *v/v* aqueous solution of acetone, firstly using a mortar and pestle, next shaken on a rocker shaker for 24 h. As acetone is not suitable for spectral measurements in the wave length range below 330 nm (because it is opaque), methanol was used instead. Therefore, extracts (2 mL) were dried under a stream of nitrogen, dissolved in 4 mL methanol and used for synchronous fluorescence spectroscopy and phenol content assessment using the spectrophotometric method with the Folin–Ciocalteu (Merck, Poland) reagent. Total free phenolics, non-tannin phenolics, total tannins, and proanthocyanidin levels were measured and calculated as described by Nasar-Abbas et al. [19] and Anonymous [20].

Synchronous Fluorescence Spectra

Synchronous spectra were measured with a Cary Eclipse spectrophotometer (Agilent, Australia) using right-angle geometry and $1 \times 1$ cm quartz cuvette. Excitation and emission slits were set at 10 nm. Spectra were registered from 240 to 720 nm with 20 nm step ($\Delta\lambda = \lambda_{em} - \lambda_{ex}$) in the range of 40–300 nm. Excitation and emission monochromator were moved simultaneously with above steps. For example, when $\lambda_{ex}$ was 240 nm then $\lambda_{em}$ was 260 nm for the first scan with $\Delta\lambda = 20$ nm and with this "window" spectrum was registered up to 720 nm for $\lambda_{em}$ wavelength. Then step was increased of 20 nm, and now was equal 40 nm and spectrum was again registered and so on. The graphs of synchronous spectra were constructed as follows: Z axis indicates steps with which spectra were registered. Colors indicating absorption bands (excitation spectrum) which cause fluorescence; therefore, in this figure there are no "nonactive absorption" and X axis shows location of absorption bands. To see which absorption bands cause fluorescence, to wavelength on X axis must be added appropriate wavelength from Z axis (in the figure $Z = \Delta\lambda$).

### 2.5. Protein Analysis

#### 2.5.1. Protein Isolation

Proteins were isolated following Rubio et al. [21] with modifications. The seed flour was defatted using a chloroform: methanol (2:1, *v/v*) mixture. After drying the material, extraction was carried out with 0.2 M borate buffer (pH 8), containing 0.5 M NaCl. The extract was subjected to ultrasounds in an ultrasonic cleaner (30 min, 4 °C) and was centrifuged ($28,000 \times g$, 30 min, 4 °C). The supernatant was recovered, its pH was adjusted with acetic acid to 4.5 and the resulting solution was shaken for 30 min. The extract was centrifuged ($28,000 \times g$, 30 min, 4 °C) and the resulting pellet was collected as legumin fraction. The supernatant was subjected to an intense dialysis against deionised water (48 h, 4 °C) and was centrifuged ($28,000 \times g$, 30 min, 4 °C). The resulting pellet (vicilin fraction) was lyophilised. Protein remaining in the supernatant was salted out from by adding ammonium sulphate $(NH_4)_2SO_4$ 608 g $\times$ L$^{-1}$, shaking the solution (2 h, 4 °C) and centrifugation ($28,000 \times g$, 30 min, 4 °C). The pellet (albumin fraction) was subjected to an intense dialysis against deionised water (48 h, 4 °C) and was lyophilised.

#### 2.5.2. One-Dimensional (1-D) Electrophoresis

Lyophilised proteins were dissolved in a buffer solution containing: 0.0625 mol $\times$ L$^{-1}$ Tris hydrochloride (HCl) (pH 6.8), 2% sodium dodecyl sulphate (SDS), 10% glycerol, and 5% 2-mercaptoethanol. The final concentration of protein was 2 mg $\times$ mL$^{-1}$. The denatured proteins were loaded to a 10% polyacrylamide gel and subjected to sodium dodecyl sulphate-polyacrylamide gel electrophoresis (SDS-PAGE) (ReadyPrep™, Bio-Rad) at 200 V for 40 min.

#### 2.5.3. Two-Dimensional (2-D) Electrophoresis

The lyophilised proteins were dissolved in a rehydration buffer ReadyPrep$^{TM}$ 2-D Starter kit (Bio-Rad, Hercules, California, USA) and loaded to 7 cm strips with a 4–7 pH gradient (IPG) strips (Bio-Rad). The strips were subjected to a passive rehydration (12 h, 20 °C) and a three stage isoelectric focusing. The preparatory stage was carried out using 250 V for 15 min, 4000 V for 2 h were used at the second stage, and 4000 V and 20,000 volt-hours were used for isoelectric focusing. The strips with separated proteins were placed on 12% polyacrylamide gel and subjected to SDS-PAGE at 200 V for 1 h. The gels were stained after protein separation with Brilliant Blue G solution (Sigma-Aldrich, Poznan, Poland) and analysed with software *PDQuest* (Bio-Rad, Hercules, California, USA). Prior to the liquid chromatography with mass spectrometry (LC-MS-MS/MS) analysis, the protein spots cut out from the gel were digested with tripsin. Spectrometric analyses were carried out in the Laboratory of Mass Spectrometry, Institute of Biochemistry and Biophysics of the Polish Academy of Science, Warsaw, Poland. The measured peptide masses were compared to the records from the Viridiplantae database using the Mascot software [22]. Protein content in solutions used for both 1-D and 2-D electrophoresis was determined with the Bradford [23] method.

### 2.6. Statistical Analysis

One-way analysis of variance (ANOVA) followed by Tukey's comparison post-hoc test ($p \leq 0.05$) was applied to evaluate differences between the storage variants: controls (temperature +20 °C) and cold storage (−14 °C) using software Statistica 6.0 (StatSoft, Krakow, Poland). At least five replicates were used for all measurements.

## 3. Results

### 3.1. Seed Viability

After 30 years of storage at −14 °C, cultivars Nadwiślański and Dino field bean (*Vicia faba* var. *minor*) seeds retained a very high germination capacity (98% and 91%, respectively). On the contrary,

the seeds stored at room temperature did not germinate at all (Table 1). Seed density was also decreased after storage. The density was lower in seeds stored at −14 °C compared to seeds stored at +20 °C in both cultivars. In plants raised from Nadwiślański seeds stored at −14 °C the average root and stem lengths were 124.26 and 45.30 mm, respectively, whereas in Dino root and stem lengths were 110.13 and 52.24 mm, respectively. In seeds stored at +20 °C seed leachate electroconductivity was four and six times higher compared to seeds stored at −14 °C in cultivars Nadwiślański and Dino, respectively.

**Table 1.** Seed viability, radicle and stem length, fresh and dry seedling mass, water content and seed leachate conductivity field bean (*Vicia faba* var. *minor*) seeds stored for 30 years at −14 °C and +20 °C.

| | Nadwiślański | | Dino | |
|---|---|---|---|---|
| | **−14 °C** | **+20 °C** | **−14 °C** | **+20 °C** |
| Germination, % | 98 ± 1A | 0 B | 91 ± 1 A | 0 B |
| Density, kg/m³ | 1379.3 ± 4.3 | 1414.7 ± 4.7 | 1373.8 ± 3.3 | 1402.1 ± 3.6 |
| Length mm: | | | | |
| Root | 124.3 ± 5.3 A | 0 B | 110.1 ± 4.6 A | 0 B |
| Stem | 45.3 ± 3.8 A | 0 B | 51.2 ± 8.7 A | 0 B |
| Seedling fresh mass, mg | 1843 ± 290A | 0 B | 1998 ± 223 A | 0 B |
| Seedling dry mass, mg | 460 ± 69 A | 0 B | 522.1 ± 39 A | 0 B |
| Water content, % | 10.6 ± 0.4 A | 6.6 ± 0.1 B | 8.6 ± 0.1 A | 6.5 ± 0.3 B |
| Electroconductivity, mS × g$^{-1}$ | 327 ± 16.5 A | 1955 ± 23 B | 416 ± 18 A | 1815 ± 20 B |
| Total protein content, % | 28.7 ± 1.4 A | 25.6 ± 1.2 B | 27.5 ± 1.2 A | 25.4 ± 1.8 B |

Note. Means denoted with different capital letters are significantly different ($p < 0.05$) across the described groups; average ± standard deviation. comparisons were made between −14 °C and +20 °C treatments within each cultivar. All differences were significant at the level 0.05.

## 3.2. Characteristics of Seed Testae

Seed storage temperature affected testae colour. Seeds stored at −14 °C were light brown, whereas those stored at room temperature were black (Figure 1).

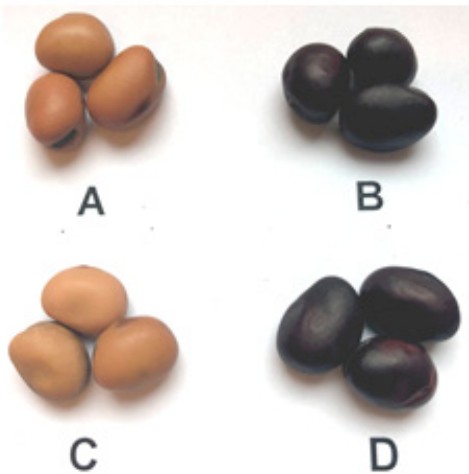

**Figure 1.** General view of stored seeds: (**A**)—Nadwiślański seeds storage at −14 °C, (**B**)—Nadwiślański seeds storage at + 20 °C, (**C**)—Dino seeds storage at −14 °C, (**D**)—Dino seeds storage at + 20 °C.

No clear structural differences between testa of various seeds were visible in SEM images, however, small differences in hilum width were noted. The hilum in seeds stored at low temperature was wider by 10 μm, compared to seeds stored at 20C (Figure 2).

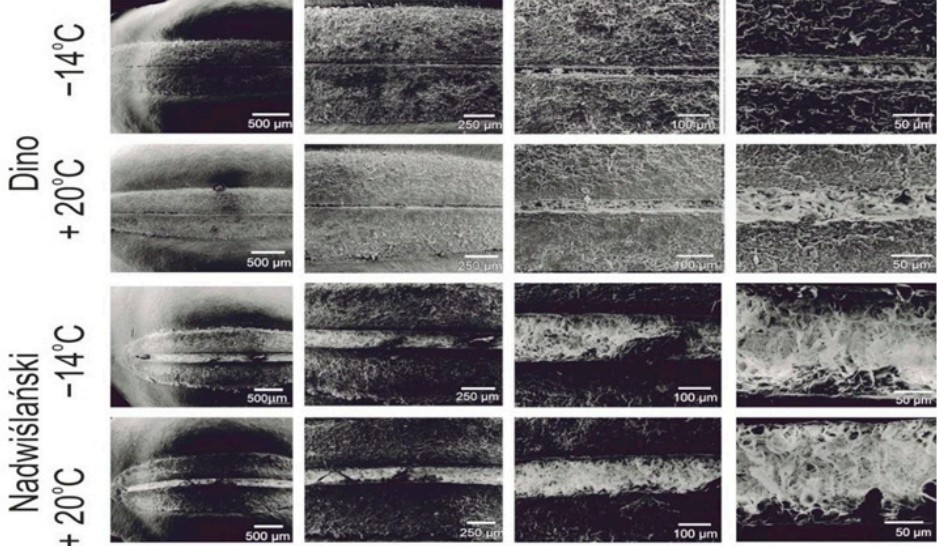

**Figure 2.** SEM images of seed hilum surface stored at different temperature. In one line are available for the same seeds at different magnifications (magnification ranging from 20× to approximately 200×, scale bar 500 to 50 μm is shown).

Synchronous fluorescence spectra were registered to characterize the biochemical changes in the seed coats (Figure 3). The measurements were carried out using the wavelengths range of 240–700 nm and $\Delta\lambda$ ($\Delta\lambda = \lambda_{em} - \lambda_{ex}$) range of 40–300 nm. Two major regions of emissions were observed with different positions and intensities depending on seed variety and storage temperature (Nadwiślański and Dino, 30 years at −14 °C and +20 °C).

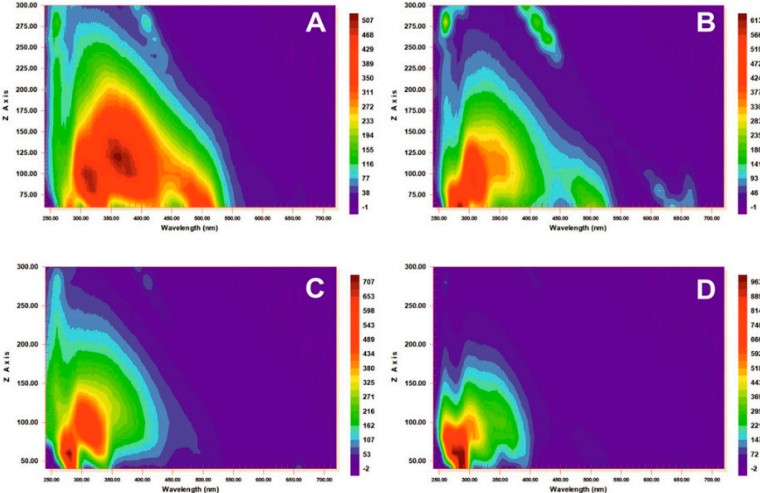

**Figure 3.** Synchronous fluorescence spectra of testa from field bean (*Vicia faba* var. *minor*) cultivars seeds stored at different temperature during 30 years: (**A**)—Nadwiślański stored at −14 °C; (**B**)—Nadwiślański stored at +20 °C; (**C**)—Dino stored at −14 °C; (**D**)—Dino stored at +20 °C. In figure Z = $\Delta\lambda$.

The maximum values of fluorescence were in the following ranges: $\Delta\lambda = 60 \div 70$ nm, $\lambda_{max} = 280 \div 290$ nm, and $\Delta\lambda = 100 \div 130$ nm and $\lambda_{max} = 320 \div 370$ nm. The spectra indicate the presence of two major fluorophore groups with similar emission properties in methanol extracts from seed coats of all samples. Comparing the emission bands of extracts from Dino seeds, analysed at $\Delta\lambda = 100$ nm different intensities were found depending on the storage regime. Measuring extracts of seeds stored at +20 °C, low intensity bands were detected within the range 250–280 nm, whereas

medium intensity bands occurred within 480–510 nm, and high intensity bands between 280 and 480 nm. On the other hand, analyzing (at the same Δλ = 100 nm) the extracts from seeds stored at −14 °C low intensity bands occurred within the ranges 250–280 nm and 330–510 nm, whereas the high intensity bands were observed located in the range of 280–330 nm.

Analysing the extracts of seed coats of the cv. "Nadwislański" the differences in fluorescence spectra corresponding to seed storage regime were also noticed, however not so clear as with the cv. Dino. Seeds stored at +20 °C were characterized by the main bands within the ranges 240–280 nm (low intensity), 380–430 nm (moderate intensity) and 280–430 nm (high intensity), whereas the extracts from seeds stored at −14 °C had their weak and moderate intensity bands within the range 250–390 nm. This observation suggests a rather complex nature of the fluorescence emission from the methanol extracts of field bean (*Vicia faba* var. *minor*) seed coats. It can be safely stated, however, that the components of the extracts have more than two major fluorophore groups. It is worth emphasizing that both the intensity of fluorescence and the full width at half maximum (FWHM) of fluorescence bands were higher (in both varieties) in the case of seeds stored at +20 °C. The spectrophotometric analysis of the composition of phenolic compounds in seed coats showed the decreased amounts of non-tannin phenolics, total tannins, and proanthocyanidins in testae of seeds stored at +20 °C (Table 2).

**Table 2.** Total free phenolic, non-tannin phenolics, total tannins, and proanthocyanidins content in field bean (*Vicia faba* var. *minor*) seeds stored for 30 years at −14 °C and +20 °C.

|  | Nadwiślański | | Dino | |
| --- | --- | --- | --- | --- |
|  | −14 °C | +20 °C | −14 °C | +20 °C |
| Total free phenolics, mg tannic acid × g$^{-1}$ | 60.64 ± 1.1 | 2.66 ± 0.2 | 62.21 ± 0.9 | 2.81 ± 0.2 |
| Non-tannin phenolics, mg tannic acid × g$^{-1}$ | 18.22 ± 1.5 | 1.54 ± 0.1 | 18.40 ± 1.7 | 1.57 ± 0.1 |
| Total tannins, mg tannic acid × g$^{-1}$ | 42.46 ± 3.5 | 1.12 ± 0,1 | 43.80 ± 2.4 | 1.24 ± 0.1 |
| Proanthocyanidins, mg leucocyanidin × g$^{-1}$ | 38.60 ± 1.9 | 5.11 ± 0.3 | 37.30 ± 2.7 | 4.80 ± 0.3 |

*Note.* Comparisons were made between −14 °C and +20 °C treatments within each cultivar. All differences were significant at the level 0.05; averages ± standard deviations are given.

### 3.3. Seed Protein Characteristics

#### 3.3.1. 1-D Electrophoresis

The patterns of polypeptide fractions (albumin, vicilin, and legumin) isolated from seeds stored at −14 °C and +20 °C distinctly differed depending on seed storage regime. Cultivar-related differences were also noted; however, they were small (Figure 4). In both field bean (*Vicia faba* var. *minor*) cultivars, most peptides had similar molecular masses, with the exception of members of the albumin fraction with masses 75.4, 43.4, 21.9, 20.5, 15.7, and 13.0 kDa, that were found in cultivar Dino only (Figure 4).

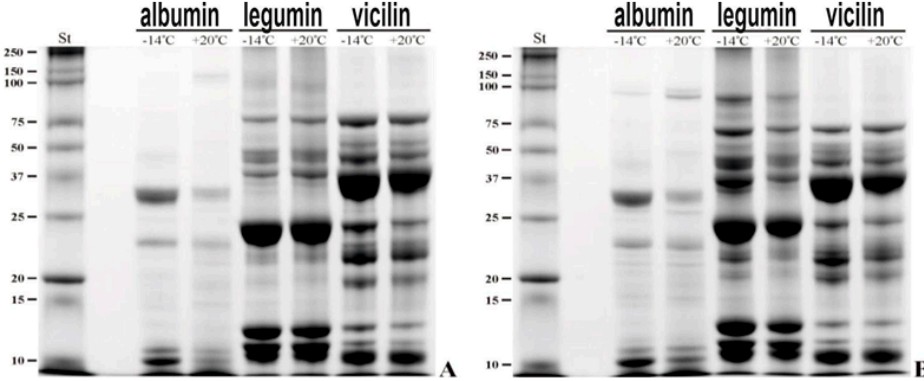

**Figure 4.** SDS-PAGE separations of albumin, vicilin, and legumin protein fractions from field bean (*Vicia faba* var. *minor*) cultivars Nadwiślański (**A**) and Dino (**B**) seeds stored at −14 °C and +20 °C (St—standard).

The albumin fraction showed the greatest differences across the studied seed lots with regard to both band numbers and intensities. Most bands were more intense in seeds subjected to cold storage, and this was particularly visible with fractions with molecular masses 30.5, 22.8, and 10.8 kDa (Figure 4). In electropherograms of seeds stored at −14 °C the intensity of the 30.5 kDa polypeptide was nearly twice as high as in protein separations of seeds stored at room temperature. The 45.2 kDa and 20.4 kDa polypeptides were not detected at all in seeds stored at +20 °C. There were three polypeptides (125.6, 105.2, and 54.1 kDa) that produced more intense bands in electropherograms of seeds stored at room temperature. Moreover, these polypeptides were not observed at all in Nadwiślański seeds stored at −14 °C (Figure 4). Within the legumin fraction most polypeptides had similar molecular masses in both cultivars. This did not hold true for the 32 kDa polypeptide that was only observed in seeds of Nadwiślański.

In Nadwiślański the electrophorograms of polypeptides from the material subjected to different storage regimes differed not only in intensity but also in the number of bands (the absence of 56.6 and 32.0 kDa peptides in seeds stored at +20 °C). In this cultivar all bands were more intense in seeds stored at −14 °C (Figure 4).

Within the vicilin fraction a 22.4 kDa peptide differentiated the two cultivars, as it was only observed in Nadwiślański (Figure 4). The intensity of bands obtained with seeds stored at room temperature differed from bands obtained with seeds stored at −14 °C. Seeds subjected to room temperature storage generally produced weaker bands, and this was particularly visible with fractions 24.1, 21.5 19.3, and 12 kDa.

### 3.3.2. 2-D Electrophoresis and Protein Identification

Plant cultivars did not have very strong effects on patterns of protein spots (from fractions albumin, vicilin, and legumin) in 2-D separations, whereas seed storage regime (−14 °C or +20 °C) affected both the number and intensity of protein spots (Figures 5–8).

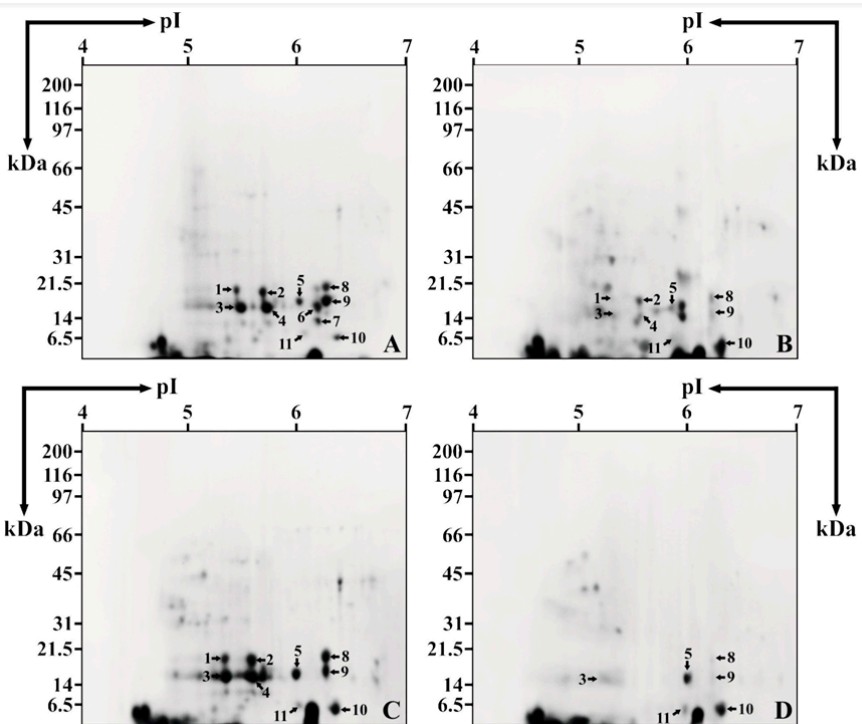

**Figure 5.** 2D-PAGE separations of the albumin fraction ((**A**)—cv. Nadwiślański seeds stored at −14 °C; (**B**)—cv. Nadwiślański seeds after +20 °C storage; (**C**)—cv. Dino seeds stored at −14 °C; (**D**)—cv. Dino seeds stored at +20 °C).

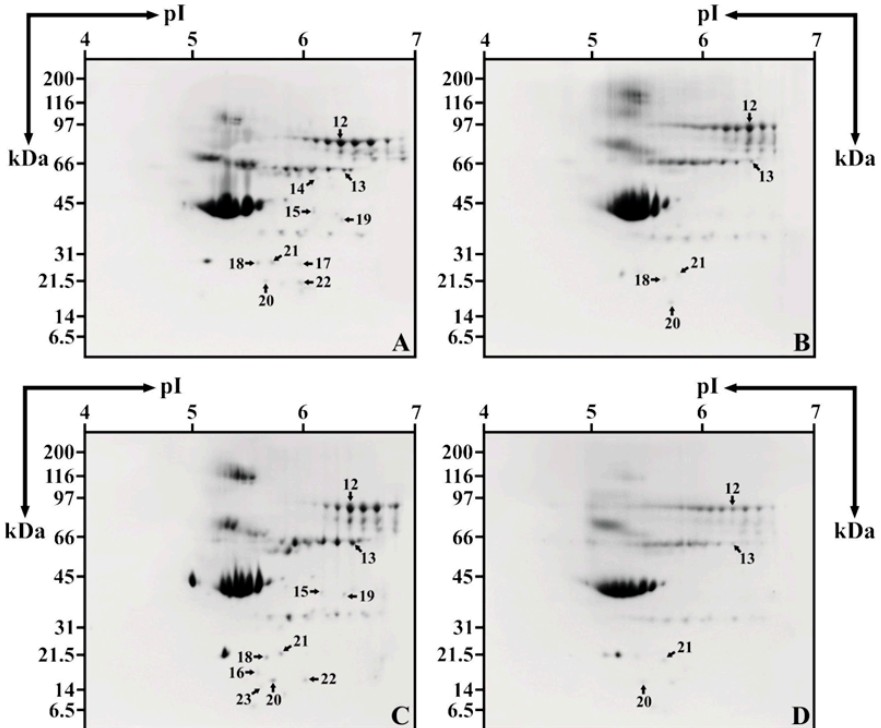

**Figure 6.** 2D-PAGE separations of the legumin fraction ((**A**)—cv. Nadwiślański seeds stored at −14 °C; (**B**)—cv. Nadwiślański seeds after +20 °C storage; (**C**)—cv. Dino seeds stored at −14 °C; (**D**)—cv. Dino seeds stored at +20 °C).

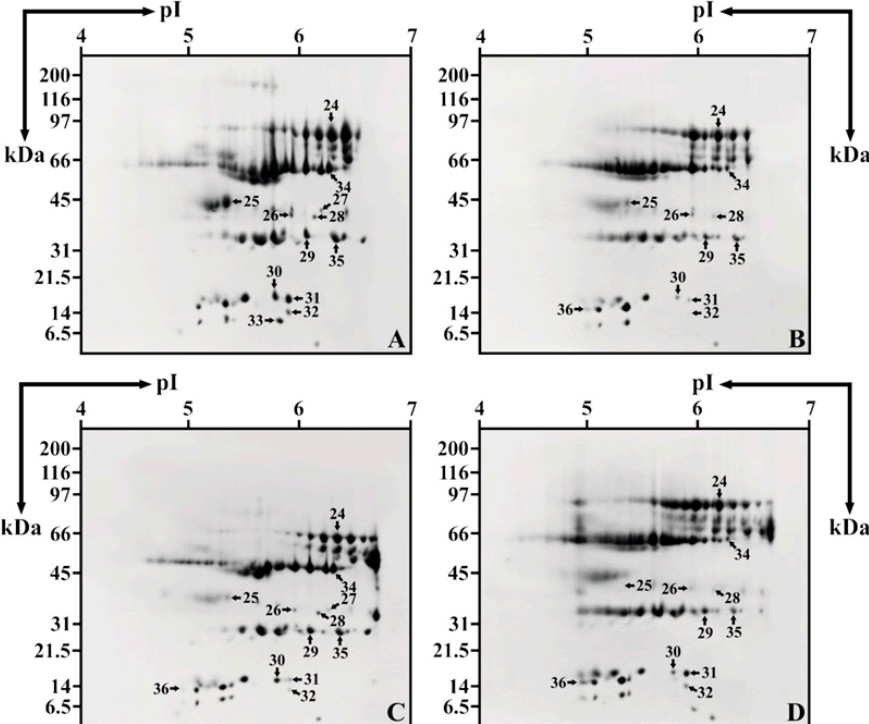

**Figure 7.** 2D-PAGE separations of the vicilin fraction fraction ((**A**)—cv. Nadwiślański seeds stored at −14 °C; (**B**)—cv. Nadwiślański seeds after +20 °C storage; (**C**)—cv. Dino seeds stored at −14 °C; (**D**)—cv. Dino seeds stored at +20 °C).

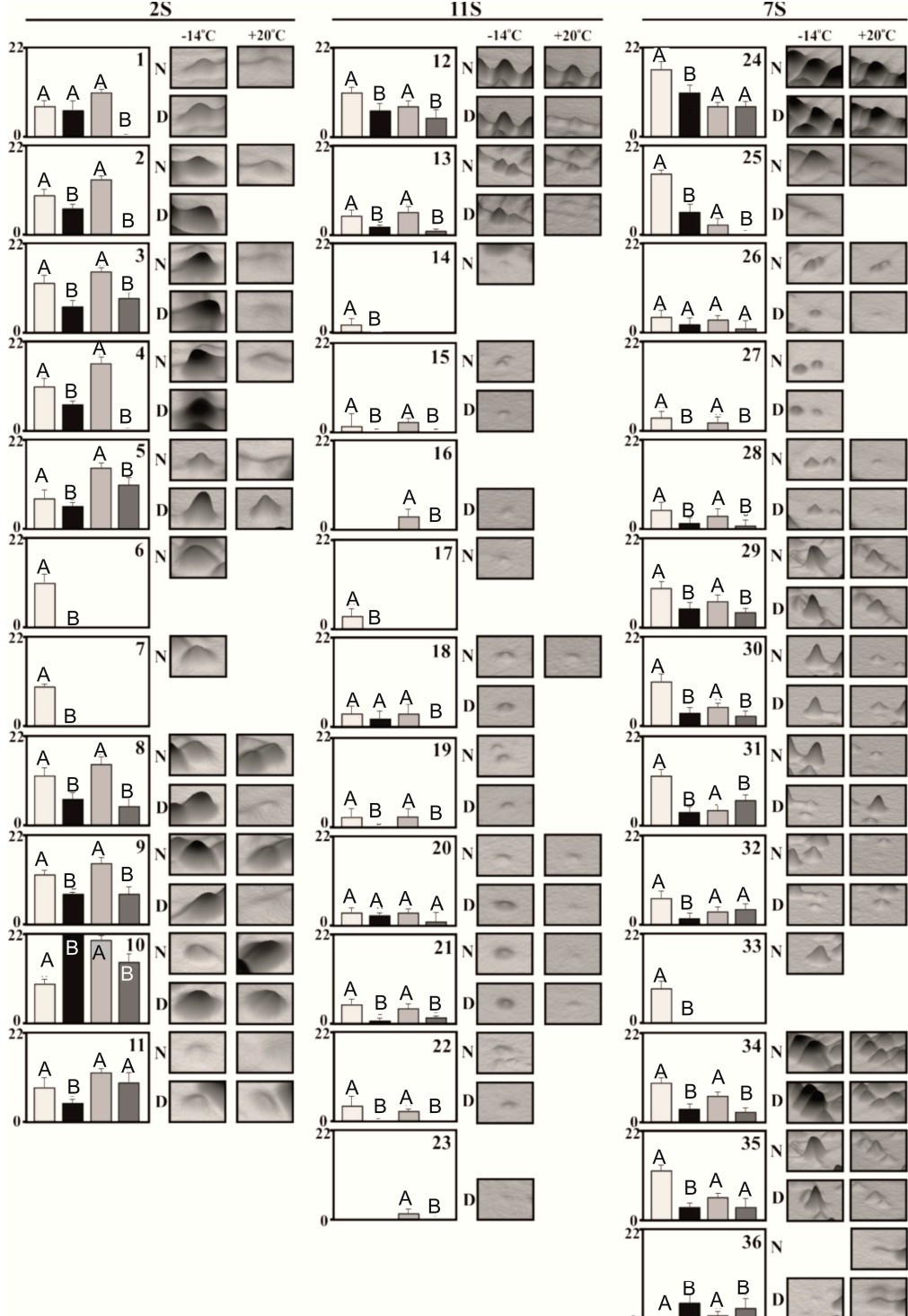

**Figure 8.** Comparison of the intensities of identified proteins (left hand panel for each fraction, albumin, vicilin, and legumin) and the three-dimensional (3-D) visualization of electrophoretic spots (right hand panel). The numbers given in each graph within the left hand panel are the numerals of identified proteins for each seed lot: Nadwiślański −14 °C □; Nadwiślański +20 °C ■; Dino −14 °C ■; Dino +20 °C ■. Data for storage temperature variants designated with the same capital letter are not significantly different ($p \leq 0.05$, Tukey's test); means labelled with different capital letters are significantly different among the described groups; bars denote standard deviation of the mean; N—Nadwiślański, D—Dino.

Albumin fraction: The 2D-PAGE maps of albumin fraction from seeds of cultivars Nadwiślański and Dino stored at −14 °C and +20 °C are shown in Figure 6. Within the whole range of molecular masses, 47 and 54 protein spots were obtained with seeds of Nadwiślański and Dino, respectively, stored at −14 °C. Seeds of the same cultivars, stored at +20 °C produced 39 and 30 spots on 2-D separations. Most polypeptides had the pI within the range 5–6.5 and the molecular masses up to 45 kDa.

Differences in the intensities of protein spots were observed with seeds stored at −14 °C vs. +20 °C. These differences were particularly visible in proteins with low molecular masses (not more than 20 kDa) and pI 5–6.5. The lack of these proteins in seeds stored at +20 °C, particularly in Dino, was clearly demonstrated. Eleven proteins, whose intensity differed most evidently, depending on seed storage regime, were subjected to mass spectrometric analyses. The results of protein identification are presented in Table 3.

**Table 3.** Albumin identification fraction by LC-MS-MS/MS.

| Spot | Protein Identification | Species | Protein ID | Molecular Function | Score |
|------|------------------------|---------|------------|-------------------|-------|
| 1 | Lectin | *Vicia faba* | P02871,1 | carbohydrate and metal ion binding | 298 |
| 2 | Lectin | *Vicia faba* | P02871,1 | carbohydrate and metal ion binding | 278 |
| 3 | Lectin | *Vicia faba* | P02871,1 | carbohydrate and metal ion binding | 382 |
| 4 | Lectin | *Vicia faba* | P02871,1 | carbohydrate and metal ion binding | 302 |
| 5 | Hypothetical protein TSUD_263810 | *Trifolium subterraneum* | GAU38157,1 | unknown | 291 |
| 6 | Vicilin | *Vicia faba* | P08438,1 | nutrient reservoir activity | 334 |
| 7 | Albumin-1C | *Pisum sativum* | P62928,1 | nutrient reservoir activity and toxin | 63 |
| 8 | Lectin | *Lathyrus sativus* | CAD27485,1 | carbohydrate and metal ion binding | 24 |
| 9 | Lectin | *Vicia faba* | P02871,1 | carbohydrate and metal ion binding | 328 |
| 10 | Superoxide dismutase (Cu–Zn) | *Pisum sativum* | Q02610,2 | destroys radicals | 474 |
| 11 | Lectin | *Vicia faba* | P02871,1 | carbohydrate and metal ion binding | 272 |

The identified proteins included the storage albumin (spot 7) and vicilin (spot 6), and defence proteins—lectins (spots 1, 2, 3, 4, 8, 9, and 11) and superoxide dismutase (spot 10). In seeds subjected to storage at +20 °C, the proteins identified as albumins, vicilins, and lectins had clearly decreased intensities or did not occur at all (Figure 8). Only one protein (superoxide dismutase) was more abundant in seeds stored at room temperature.

Legumin fraction: 2-D protein maps obtained with seeds stored at −14 °C contained on average 82 and 75 spots in cultivars Nadwiślański and Dino, respectively, whereas with seeds of the same cultivars, stored at +20 °C, 67 and 63 spots were obtained (Figure 6). The pI of most polypeptides was within the range 5–7.8 and the molecular masses 97–20 kDa. Seeds stored at room temperature did not contain polypeptides with molecular masses 52–38 kDa and pI 5.8–6.5, as well as polypeptides with molecular masses 32–18 kDa and pI 5.5–6.1. Twelve proteins were identified. Six of them showed similarity to legumin (spots 16, 17, 18, 19, 20, and 22), one to convicilin (spot 12) and five to vicilin (spots 13, 14, 15, 21, and 23) (Table 4).

**Table 4.** Legumin identification fraction by LC-MS-MS/MS.

| Spot | Protein Identification | Species | Protein ID | Score |
|------|------------------------|---------|------------|-------|
| 12 | Convicilin | *Vicia faba* | CAP06335,1 | 2219 |
| 13 | Vicilin | *Vicia faba* | P08438,1 | 2304 |
| 14 | Vicilin | *Vicia faba* | P08438,1 | 1231 |
| 15 | Vicilin | *Vicia faba* | P08438,1 | 1425 |
| 16 | Legumin propolypeptide alpha chain | Papilionoideae | AAB24084,1 | 1037 |
| 17 | Legumin A2 primary translation product | *Vicia faba* var. *minor* | CAA38758,1 | 1125 |
| 18 | Legumin type B | *Vicia faba* | P05190,1 | 504 |
| 19 | Legumin A2 primary translation product | *Vicia faba* var. *minor* | CAA38758,1 | 717 |
| 20 | Legumin propolypeptide beta chain | Papilionoideae | AAB24085,1 | 360 |
| 21 | Vicilin | *Pisum sativum* | P13918,2 | 533 |
| 22 | Legumin A2 primary translation product | *Vicia faba* var. *minor* | CAA38758,1 | 560 |
| 23 | Vicilin | *Vicia faba* | P08438,1 | 1003 |

Two identified proteins (spots 14 and 17) were specific for Nadwiślański and two others (spots 16 and 23)—for Dino (Figures 6 and 7). All identified proteins were more abundant in seeds stored at

−14 °C (Figure 8), and they all could be related to the same molecular function—nutrient reservoir activity. Five identified proteins could not be detected in Nadwiślański seeds stored at + 20 °C and six identified proteins were absent from Dino seeds stored at room temperature.

Vicilin fraction: In seeds stored at −14 °C this fraction produced 93 and 72 protein spots in cultivars Nadwiślański and Dino, respectively. In seeds of the same cultivars, stored at room temperature these proteins produced 65 and 66 spots (Figure 7). The molecular masses of the majority of these proteins were in the range 97–10 kDa and pI 4.8–6.7. Differences were observed between seeds stored at −14 °C and +20 °C, regarding protein spot intensity, particularly in the case of proteins with molecular masses 50–30 kDa and pI 5–6.7. The identification of thirteen of these proteins was carried out (Table 5).

**Table 5.** Vicilin identification by LC-MS-MS/MS.

| Spot | Protein Identification | Species | Protein ID | Score |
|------|------------------------|---------|------------|-------|
| 24 | Convicilin | *Vicia faba* | CAP06335,1 | 2060 |
| 25 | Legumin A2 primary translation product | *Vicia faba* var. minor | CAA38758,1 | 1441 |
| 26 | Vicilin | *Vicia faba* | P08438,1 | 771 |
| 27 | Vicilin | *Vicia faba* | P08438,1 | 1206 |
| 28 | Vicilin | *Vicia faba* | P08438,1 | 1358 |
| 29 | Vicilin | *Vicia faba* | P08438,1 | 1832 |
| 30 | Vicilin | *Vicia faba* | P08438,1 | 575 |
| 31 | Vicilin | *Vicia faba* | P08438,1 | 637 |
| 32 | Vicilin | *Vicia norbonensis* | CAA96514,1 | 270 |
| 33 | Vicilin | *Vicia faba* | P02871,1 | 216 |
| 34 | Vicilin | *Vicia faba* | P08438,1 | 1126 |
| 35 | Vicilin | *Vicia faba* var. minor | CAA68525,1 | 639 |
| 36 | Vicilin | *Vicia faba* | P08438,1 | 867 |

There were eleven vicilin-related proteins (spots 26, 27, 28, 29, 30, 31, 32, 33, 34, 35, and 36) one convicilin-related (spot 24) and one legumin-related (spot 25). One vicilin-related protein (spot 33) was unique for Nadwiślański (Figures 7 and 8). Ten of the identified proteins were more abundant in sees stored at −14 °C. In Nadwiślański two proteins were found (spots 27 and 33) that were absent in seeds subjected to room temperature storage, and there was one such identified protein (spot 27) in Dino. Two vicilin-related proteins (spots 32 and 33) were more abundant in Nadwiślański seeds stored at −14 °C but were less abundant in Dino seeds subjected to cold storage, compared to such seeds stored at room temperature. All identified vicilin were related to the nutrient reservoir activity.

## 4. Discussion

### *4.1. Vigour and Viability*

Temperature and air humidity are considered the main factors affecting seed viability [24]. Field bean (*Vicia faba* var. *minor*) seeds that were dried and stored for 30 years at −14 °C germinated at 91% and 98% in cultivars Dino and Nadwiślański, respectively. Seed storage at room temperature resulted in a complete loss of germination capability (Table 1). Our studies thus confirm that low temperature storage of seeds favours the preservation of seed viability. Similar observations were made by Pradhan and Badola [25]. Lee et al. [26] compared viability of seeds belonging to selected species in the Fabaceae family and found that it decreased by 15% in pea and soybean seeds as a result of a 10-year storage at +4 °C. Walters et al. [27] reported various degrees of viability loss in seeds of Fabaceae (47 species) stored at −18 °C. For instance, within the *Trifolium* genus *Triffolium campestre* did not lose viability after 44-year storage, whereas *T. caudatum* after 38 years of storage had the germination capacity of just 2%. In *Vicia* sp. seeds stored for 42 years the germination percentage decreased by just 20%.

An important prerequisite for preservation of high seed quality is to keep the air relative humidity and seed water content at a low level [9]. Mira et al. [28], however, maintain that seed water content does not affect seed viability if only the storage temperature is below +30 °C. In our experiments seed water content was 8.8% at the start of the experiments and in Nadwiślański after 30 years of storage

at −14 °C and +20 °C it shifted to 10.6% and 6.7%, respectively. Similarly, in Dino, after the same period of storage at −14 °C and +20 °C the seed water content changed to 8.6% and 6.5%, respectively (Table 1). In both cultivars (Dino and Nadwiślański), therefore, seeds stored at room temperature had a lower water content than those stored at −14 °C. Pérez-García et al. [29] have also found that changes in seed water content within the range 2–15% do not affect seed lifespan under the conditions of low temperature. Considering these data, we assumed that the air relative humidity and seed water content did not affect field bean (*Vicia faba* var. *minor*) seed lifespan in our experiments.

### 4.2. Seed Coat Characteristics

Phenolic compounds affect the plant metabolic activity and antioxidative properties of plant derived foods [30,31]. They also contribute to the colour of plant products [32]. In synchronous fluorescence method the intensity of fluorescence is measured as a function of the emission and excitation wavelengths with difference step of $\Delta\lambda$ or $\Delta\nu$. Good resolution and multiplying of absorption and emission intensities (in comparison to the conventional studies of emission spectra) are the major advantages of this method [33,34]. The Stokes shifts were published corresponding to the best $\Delta\lambda$ for observations of synchronous spectra of various phenols. Chlorogenic acid in a methanol solution, for instance, has a single fluorescence maximum 337 nm for $\Delta\lambda = 110$ nm [35]. Sergiel et al. [36] studied several phenolics dissolved in methanol and gave the wavelength ranges and $\Delta\lambda$ values suitable for the analyses of these compounds (Table 6).

**Table 6.** The emission and excitation wavelengths of selected phenols [36].

| Phenolic Acids | $\lambda_{ex}$, nm | $\Delta\lambda$, nm |
|---|---|---|
| Caffeic acid | 365–375 | 80–95 |
| Chlorogenic acid | 355–365 | 90–110 |
| 2,5-Dihydroxybenzoic acid | 305–325 | 130–150 |
| Ferulic acid | 355–365 | 60–95 |
| Gallic acid | 305–315 | 60–90 |
| Homogentisic acid | 280–290 | 50–60 |
| 4-Hydroxybenzoic acid | 275–280 | 40–60 |
| Vanillic acid | 260–280 | 60–95 |
| Syringic acid | 260–280 | 85–105 |

Considering the data given in Table 6 and the obtained results, we can assume that the seed coats of field bean (*Vicia faba* var. *minor*) may contain all of phenolic metabolites mentioned there. To gain a better insight into the changes occurring in seed coats during prolonged storage, the total content of phenols, non-tannin phenolics, total tannins, and proanthocyanidins were measured. The obtained results show clearly that in seeds damaged by inadequate storage conditions the contents of all the above-mentioned compounds decreases, by up to 95% (Table 2). On the other hand, the synchronous fluorescence spectra show increased levels of fluorescent metabolites in seeds stored at +20 °C (increased band intensity, Figure 3). We should remember that phenols in spite of their wide structural diversity share the propensity for becoming oxidized [37]. Nasar-Abbas et al. [19] demonstrated, that in bean a 12-month storage in the atmosphere of oxygen results in accelerated seed darkening, whereas the atmosphere of nitrogen delays this change. A condensed tannin contributes to the darkening of pinto bean seed coats and this process occurs more rapidly in a cultivar with a higher initial content of proanthocyanidin compared to the one with a low content of this metabolite [38]. Seeds of pinto bean contain the main monomer of the flavonol, kaempferol, and three flavonols—kaempferol 3-*O*-glucoside, kaempferol 3-*O*-glucosylxylose, and kaempferol 3-*O*-acetylglucoside. Seed ageing resulted in a decrease of kaempferol content by nearly a half and it did not change significantly in those seeds that did not change their colour. In seed coats procyanidins are the main polymers with the degree of polymerisation above 10. Seed ageing decreased the level of these polymers and increased the level of low molecular weight procyanidins. However, our results do not show the occurrence of

free kaempferol in field bean (*Vicia faba* var. *minor*) seeds (Figure 3) (the extract spectra do not contain the bands corresponding to kaempferol). The most significant features of phenolics are their ability to undergo oxidation and to polymerise. The decreased contents of total phenols and non-tannin phenols may probably result from dimerization of polyphenols leading to the formation of insoluble high molecular weight polymers or products of their oxidation. Two reasons of the browning of lentil seeds were described: polymerisation of phenols [39] and oxidation of phenols [40]. Our results, the observation of increased fluorescence in methanol extracts from field bean (*Vicia faba* var. *minor*) seeds stored at +20 °C, thus indicate that oxidized forms of phenols or condensed phenols can be formed in seed coats of such inadequately stored seeds.

### 4.3. Characteristics of Protein

Although there have been numerous reports on field bean (*Vicia faba* var. *minor*) seed storage (e.g., [41,42]) and the storage proteins of *Vicia faba* (e.g., [43–45]), there have been few reports on the relation between seed storage proteins and the storability of grain legume seeds. Mbofung et al. [46] did not notice any correlation between total protein content and vigour and viability of soybean seeds; however, Dobiesz and Piotrowicz-Cieślak [47] indicate that in yellow lupin there is a correlation between the levels of conglutins γ and δ and seed biological quality. Simiarly, Sathish et al. [48] described a close relation of some proteins from black gram seeds to seed viability and vigour.

Shaban [49] found that various factors may cause both quantitative and qualitative changes in seed storage proteins. Proteins make a large part of legume seed dry mass—from approximately 20% in pea and bean to 38–40% in soybean and lupin. Field bean (*Vicia faba* var. *minor*) seeds used in our experiments contained proteins at the level of up to 26.8% of their dry mass (Table 1) which corroborates the data provided by Księżak et al. [50]. Considering the differences in their solubility, seed proteins are divided into four groups: albumins (soluble in water and buffered solutions with neutral acidity), globulins (soluble in salt solutions), prolamins (soluble in alcohols), and glutelins (soluble in acids and alkali) [51]. In our experiments seed storage proteins were divided into the following fractions: albumin, vicilin, and legumin, based on solubility controlled by pH and salt concentration, following the method of Rubio et al. [19]. Although the designations of these fractions suggest large differences in molecular weights of these proteins, their electrophorograms contain diverse peptides with size ranges partially overlapping. This may be explained by the fact, well established that seed storage proteins have an oligomeric structure and electrophoresis reveals their subunits [52,53]. Non-germinating seeds were distinguished in our work by decreased intensity of electrophoretic bands and, with some proteins, by their complete disappearance (Figures 5–8). The largest protein differences related to seed storage regime were observed within the albumin fraction. Polypeptides with molecular masses 45.2 and 20.4 kDa were not detected in nongerminating seeds. Similarly, within the legumin fraction of Nadwiślański there was seed protein bands corresponding to molecular masses 56.6 and 32 kDa that were not observed in the deteriorated, non-germinating seeds. The relationship between seed vigour/viability and seed proteins was also reported by Mbofung et al. [46] and Sathish et al. [48]. The disappearance of protein bands from ageing lupin seeds was noted by Dobiesz et al. [22]. Rajjou et al. [6] and Sathish et al. [48] suggested that the disappearance of some protein bands results from protein degradation in stored seeds. Gao et al. [54] found that the content of seed storage proteins may be the main determinant of seed viability. 2-D electrophoresis of pre-purified fractions enabled us to identify proteins, the occurrence of which was most clearly related to the seed storage regime.

The use of 2-D electrophoresis to study seed proteins can be a challenge. Storage proteins usually greatly prevail in the separations and mask the members of other groups and functional categories. Seeds of Nadwiślański and Dino, stored at −14 °C were characterised by 47 and 54 spots, respectively, within the albumin fraction (Figure 3). The numbers of albumin spots obtained with seeds stored at +20 °C were lower by 8 and 24 in Nadwiślański and Dino, respectively. The albumins occurring in large quantities in seeds of many crops, e.g., soybean, sunflower, mustard, or Brazil nut, where they fulfil the storage functions [55]. Albumins of some plants have already been subjected to in

depth analyses, e.g., in common castor (*Ricinus communis*)—RicC1 and RicC3, common sunflower (*Helianthus annuus*)—SFAs (sunflower albumins), soybean (*Glycine max*)—AL1 and AL3 proteins, or garden pea (*Pisum sativum*)—PA2 proteins. Albumin-like storage proteins are even accumulated by some ferns; for instance, matteucin protein occurs in *Matteuccia struthiopteris* spores [51]. In this paper among 11 identified proteins belonging to the albumin fraction and related to seed storage regime, seven turned out to be lectins (Table 3). Their abundance decreased in seeds stored at room temperature (spots 1, 2, 3, 4, 8, 9, and 11) (Figure 8) or they totally disappeared from such seeds (spots 1, 2, and 4 in Dino). Lectins can have antimycotic properties; however, they are not able to bind the glycoconjugates in fungal cell membranes or penetrate the fungal cells due to impermeable fungal cell walls. Their action may, therefore, be limited to binding the carbohydrates on fungal cell wall surfaces, and thus hindering the synthesis or deposition of chitin [56]. The decrease in seed lectin content is likely, therefore, to promote seed infection by fungi.

Superoxide dismutase (Cu–Zn) was identified on 2-D proteomic maps as spot number 10 (Figure 8). The content of superoxide dismutase protein was higher in Nadwiślański seeds stored at +20 °C compared to seeds stored at −14 °C. A similar pattern, however regarding enzyme activity, not enzymatic protein quantity, was observed in *Jatropha curcas* L. seeds, where the degree of seed deterioration correlated with an increase in superoxide dismutase activity after 15 months storage [57].

On the other hand, Sahua et al. [58] suggests that expression of a specific superoxide dismutase isoenzyme is positively correlated with seed viability. It should be emphasized that plant superoxide dismutases can be classified according to their different cofactors, Cu–Zn, Fe, or Mn. Cu–Zn superoxide dismutase can be found in the cytosol, chloroplasts, and peroxisomes, while Fe superoxide dismutase is mainly found in chloroplasts, to a lesser extent in peroxisomes and apoplasts, and Mn superoxide dismutase is mainly found in mitochondria [59]. Thus, the determination of overall superoxide dismutase activity does not take into consideration the occurrence of several isoforms of this enzyme with potentially different activation patterns.

Globulins are even more abundant in plants than the albumins. They include, e.g., vicilin, convicilin, and legumin of *Vicia faba* and *Pisum sativum*, conglycynin and glycynin of *Glycine max*, or phaseolin of *Phaseolus vulgaris* [60]. Storage proteins of *V. faba* include mainly the globulins and can be classified into two groups with different sedimentation constants: legumins and vicilins. The 2-D electrophoretic maps obtained for the legumin fraction contained 82 and 75 spots in the case of seeds of Nadwiślański and Dino, respectively, stored at −14 °C. In seeds stored at +20 °C, 67 and 63 spots were obtained in Nadwiślański and Dino, respectively (Figure 8). Identification of the legumin revealed 6 legumins (spots 16, 17, 18, 19, 20, and 22), a convicilin (spot 12) and five vicilins (spots 13, 14, 15, 21, and 23). The vicilin fraction produced 93 and 72 protein spots in the case of Nadwiślański and Dino seeds, respectively, stored at −14 °C. The vicilin fraction was separated into 65 and 66 spots in the case of seeds of the same cultivars stored at +20 °C (Figure 8). The changes in vicilin-related protein contents were observed in ageing maize (*Zea mays* L.) seeds [61], and the rapid decline of these proteins correlated with decreasing germination rate and seedling growth rate.

## 5. Conclusions

In our experiments we observed decrease and disappearance of vicilins in deteriorated field bean (Vicia faba var. minor) (*Vicia faba* var. *minor*, Nadwiślański and Dino) seeds. It seems therefore, that the role of storage proteins is not limited to providing nutrients during germination, as they are also important for preservation of seed vigour and viability, necessary to even start the germination process. We also demonstrated the decrease of low molecular weight phenolic metabolites contents in testae of ageing field bean (Vicia faba var. minor) seeds, indicating that oxidized forms of phenols or condensed phenols can be formed in seed coats of such inadequately stored seeds. With proper storage conditions (−14 °C as opposed to 20 °C) field bean (*Vicia faba* var. *minor*) seeds can retain their high vigour for 30 years. The deterioration of seeds stored at 20 °C can be shown by changes in their colour, condensed phenol contents, storage proteins, and seed leachate electroconductivity. 2-D protein

electrophoresis and synchronous fluorescence measurements may be included in seed quality tests along more traditional methods like seed leachate electroconductivity or germination tests.

**Author Contributions:** Conceptualization, A.I.P.-C.; methodology, D.J.M., B.S., H.G., W.P., and K.G.; software, H.G.; validation D.J.M. and B.S.; formal analysis, D.J.M., M.K., B.S., H.G., W.P., and K.G.; visualization, A.I.P.-C., B.S., and K.G.; supervision, A.I.P.-C. All authors have read and agreed to the published version of the manuscript.

**Funding:** This research received no external funding.

**Conflicts of Interest:** The authors declare no conflict of interest.

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
