# Peer review of "Physiological Characteristics of Field Bean Seeds (Vicia faba var. minor) Subjected to 30 Years of Storage"

_agriculture, doi:10.3390/agriculture10110545_

Round 1

Reviewer 1 Report

I read with interest the paper entitled: "Physiological Characteristics of Field Bean Seeds Subjected to 30 Years of Storage". Seed longevity is a plant trait of fundamental importance for genetic resources conservation and therefore for plant breeding and food security. Relatively few studies have dealt with the physiological characterization of seed longevity, therefore I have appreciated the approach followed in this paper. Here below the observations I have on this manuscript: 

  • Line 33: I would say conserved or stored rather than “collected”,
  • Line 33: Insert the scientific name,
  • Line 60: proteins repeated 2 times,
  • Line 84: according to the rules of botanical nomenclature the name of the cultivars should be inserted in single quotes, i.e. ‘NadwiÅ›laÅ„ski’ and ‘Dino’.
  • Lines 98-105: I do not see mentions at replicates during the viability and seedling growth experiment, were replications performed?
  • Lines 176-179: Were Anova assumptions always met?
  • Line 210: please insert numerical value, were those differences significant?
  • Line 315: Table 3 is missing
  • General comment: throughout the results section several differences are highlighted but there are no F values, degree of freedoms and more importantly p-values; how can the reader know if those differences are indeed statistically different?
  • Figure 8: the letters and therefore significances are barely unreadable.
  • Line 388: Strange that the MC changed so much during conservation in hermetic containers, do you have an explanation for that?
  • Table 6: You should not put a table in the discussion section, you can just report these data in the text.
  • Line 431: remove “However”.
  • Conclusions: I think that the authors should discuss in more detail, the implications that their results may have in improving seed conservation.

Reviewer 2 Report

Dear authors,

I find the article very interesting, especially the part discussing proteomic analysis of the seeds. My observations are embedded in the manuscript.
